# Dysbiosis Triggers ACF Development in Genetically Predisposed Subjects

**DOI:** 10.3390/cancers13020283

**Published:** 2021-01-14

**Authors:** Stefania De Santis, Marina Liso, Mirco Vacca, Giulio Verna, Elisabetta Cavalcanti, Sergio Coletta, Francesco Maria Calabrese, Rajaraman Eri, Antonio Lippolis, Raffaele Armentano, Mauro Mastronardi, Maria De Angelis, Marcello Chieppa

**Affiliations:** 1Department of Pharmacy-Drug Science, University of Bari Aldo Moro, 70126 Bari, Italy; stefania.desantis@uniba.it; 2Research Department, National Institute of Gastroenterology “S. de Bellis”, Research Hospital, 70013 Castellana Grotte, Italy; marinaliso@libero.it (M.L.); elisabetta.cavalcanti@irccsdebellis.it (E.C.); sergiofalaut@hotmail.it (S.C.); antonio.lippolis@irccsdebellis.it (A.L.); raffaele.armentano@irccsdebellis.it (R.A.); mauro.mastronardi@irccsdebellis.it (M.M.); 3Department of Soil, Plant and Food Sciences, University of Bari, 70126 Bari, Italy; mirco.vacca@uniba.it (M.V.); francesco.calabrese@uniba.it (F.M.C.); 4Department of Pharmacy, University of Salerno, 84084 Fisciano, Italy; gverna@unisa.it; 5School of Health Sciences, College of Health and Medicine, University of Tasmania, Launceston, TAS 7250, Australia; rajaraman.eri@utas.edu.au

**Keywords:** colorectal cancer, microbiota, dysbiosis, ulcerative colitis, murine model

## Abstract

**Simple Summary:**

Colorectal cancer (CRC) is a non-communicable disease resulting from the combination of a genetic predisposition and environmental triggers, possibly including intestinal microbiota composition. However, the relationship between microbiota modulation and other CRC risk factors is still debated. With the intent to shed light on the axis between microbial imbalance and ACF (aberrant crypt foci) development in genetically predisposed individuals, we used the Winnie-APC^Min/+^ model combining genetics and inflammation. Our results indicate that the mother’s microbial composition can be a transmittable risk factor favoring the cascade of events finally resulting in ACF development in the offspring. In light of these results, preventive strategies developed to avoid dysbiosis could help to reduce the risk of tumor lesion onset and progression. These preventive approaches may be particularly effective during pregnancy and lactation to reduce a child’s risk of CRC development.

**Abstract:**

Background: Colorectal cancer (CRC) is the third most common cancer worldwide, characterized by a multifactorial etiology including genetics, lifestyle, and environmental factors including microbiota composition. To address the role of microbial modulation in CRC, we used our recently established mouse model (the Winnie-APC^Min/+^) combining inflammation and genetics. Methods: Gut microbiota profiling was performed on 8-week-old Winnie-APC^Min/+^ mice and their littermates by 16S rDNA gene amplicon sequencing. Moreover, to study the impact of dysbiosis induced by the mother’s genetics in ACF development, the large intestines of APC^Min/+^ mice born from wild type mice were investigated by histological analysis at 8 weeks. Results: ACF development in 8-week-old Winnie-APC^Min/+^ mice was triggered by dysbiosis. Specifically, the onset of ACF in genetically predisposed mice may result from dysbiotic signatures in the gastrointestinal tract of the breeders. Additionally, fecal transplant from Winnie donors to APC^Min/+^ hosts leads to an increased rate of ACF development. Conclusions: The characterization of microbiota profiling supporting CRC development in genetically predisposed mice could help to design therapeutic strategies to prevent dysbiosis. The application of these strategies in mothers during pregnancy and lactation could also reduce the CRC risk in the offspring.

## 1. Introduction

Colorectal cancer (CRC) is the third most common cancer with an incidence and mortality rate rapidly increasing worldwide, particularly in industrialized countries [1]. Due to the multifactorial nature of CRC, several risk factors have been identified, i.e., genetics, environment, and lifestyle [2]. Overall, these risk factors result in an increased chronic inflammatory state leading to cancer development, in accordance with the inclusion of inflammation as the seventh hallmark feature of cancer [3]. Therefore, numerous studies have established the negative impact of a sedentary lifestyle and dietary regimens enriched in processed foods and animal fat in association with a low fiber intake [4]. Moreover, the beneficial role of healthy compounds from fruits and vegetables is well established for intestinal disorders [5,6] as well as for other inflammatory conditions [7,8,9].

Increasing data demonstrate that all CRC risk factors, including diet, could modulate the gut microbiota, a community of microorganisms (i.e., bacteria, virus, fungi, archaea, and protist) colonizing our gastrointestinal tract [10]. This complex ecosystem is considered an active organ able to interact with the host and which is directly involved in various processes, such as tissue development, nutritional absorption, metabolism, and immunity [11]. Imbalance in the gut microbiome, called dysbiosis, is associated with numerous human diseases, including cancer [12,13]. Hence, the microbiome involvement in the pathogenesis of cancer has also been recognized for CRC [14,15]. It is widely accepted that CRC patients are depleted in bacteria producing short chain fatty acids (SCFAs), metabolites that contribute to preserving intestinal homeostasis [16,17]. Moreover, in CRC patients an enrichment in pro-inflammatory bacterial taxa, as well as those producing toxins, has been observed, causing direct damage to gut cells. Among these, some strains could produce enterotoxins that, in combination with reactive oxygen species, cause oxidative DNA damage, epithelial barrier disruption, and inflammation [18,19]. In CRC development, the β-catenin signaling plays a pivotal role in inducing cellular proliferation. In this line, different bacterial species have been found to be able to activate β-catenin signaling (e.g., *Bacteroides fragilis* and *Fusobacterium nucleatum*) [20,21]. Moreover, an increase in cellular proliferation could also be related to the genotoxin colibactin produced by the pks+ *E. coli* strains reporting the polyketide synthase gene complex (pks) gene [22]. On the contrary, the decrease in SCFA detection is mainly related to low abundances (up to a lack) of butyrogenic bacteria, such as *Lachnospiraceae*, *Ruminococcaceae*, *Bifidobacteriaceae*, *Lactobacillaceae*, and specific species, such as *Faecalibacterium prausnitzii* [23,24,25,26].

However, the microbiota profile in CRC is not universally valid, suggesting a synergistic action of both harmful and protective bacteria in colonic tumor initiation and/or progression. This synergy is also supported by the “driver–passenger theory” in which bacterial drivers (indigenous intestinal bacteria) act as first triggers that promote cancer initiation [27]. Later, the ongoing tumorigenesis induces changes in the surrounding microenvironment, allowing a competitive advantage for opportunistic bacteria (defined as bacterial passengers) able to enforce tumor progression. The “driver–passenger theory” suggests that bacterial drivers and passengers have distinct temporal associations with CRC development, and therefore they could have distinct roles in CRC pathogenesis [27].

In general, several studies report a different composition in terms of gut microbiota between stool samples from CRC patients and control subjects, as well as from tumor and adjacent non-tumor mucosa in patients with CRC [28,29]. Moreover, changes in gut microbiota have been reported for all stages of CRC development from adenomatous polyps to early-stage cancer to metastatic disease [30,31,32]. This could be particularly important considering a hypothetical use of specific microbial patterns as biomarkers for an early detection of CRC, with the aim to improve screening strategies. In addition, gut microbiota can also influence the efficacy or toxicity of therapy in CRC [33]. In this line, considering the promoting and supporting role of gut microbiota in CRC tumorigenesis, several strategies able to specifically target pro-tumoral bacteria have been developed. Among these, antibiotics indirectly affect CRC progression modulating the gut microbiota, even if their use generally has broad effects facilitating the acquisition of drug resistance. In addition, fecal microbiota transplantation (FMT), first described for *Clostridium difficile* infection (CDI) treatment [34], stands out as a promising strategy in the treatment of CRC patients, even if potential risks such as the selection of donors and recipients and the establishment of a tolerable dose and frequency are still debated concerning FMT in CRC, as well as in other diseases [35].

The actual knowledge about gut microbiota modulation in CRC tumorigenesis derives from numerous clinical and preclinical studies. Specifically, in genetic mouse models of CRC, such as APC^Min/+^ mice [36,37], as well as in the commonly used model of colitis-associated CRC (CAC), i.e., the AOM/DSS model [38,39], the role of bacteria in the initiation and promotion of CRC has been extensively demonstrated. However, a CRC mouse model giving an integrated view on the impact of genetics and inflammation in terms of gut microbiota modulation is still missing.

For this reason, taking advantage of our recently established mouse model (the Winnie-APC^Min/+^ model) [40] that combines both an inflammatory background (Winnie mice) [41] and a genetic predisposition to develop small intestinal polyposis (APC^Min/+^ mice), in the present study we performed gut microbiota profiling to identify bacterial patterns supporting CRC tumorigenesis in genetically predisposed mice. In fact, Winnie mice, due to a single missense mutation (G9492A, GenBank accession no. AJ511872) in the Mucin 2 (Muc2) gene, can be considered as a low-grade, spontaneous, and progressive model of ulcerative colitis (UC) [41,42]. Moreover, we aim to study the genetics impact of Winnie^+/−^ breeders dictating a dysbiotic state on the aberrant crypt foci (ACF) development in APC^Min/+^ puppies. This concept may introduce the paradigm of CRC as a communicable disease that spreads from mothers to child via intestinal microbiota transfer.

## 2. Results

### 2.1. Intestinal Microbiota in Winnie-APC^Min/+^ Mice Shows Dysbiotic Features Typical of Winnie Mice

To identify the microbial profiling promoting CRC development, we evaluated the microbiota of Winnie-APC^Min/+^ mice and their littermates obtained from breeders in a heterozygous state for the Winnie mutation (for male and female) plus the APC^Min/+^ mutation (only for male) [40]. Gene amplicon sequencing analysis of fecal material collected from 8-week-old mice reported 28,301.18 ± 17,409.02 number of reads per sample. The percentage 89.09 ± 2.12 of total reads per sample was assigned at least to the genus level. Microbial differences among Winnie-APC^Min/+^ mice and their littermates have been evaluated. Analysis of α-diversity showed significant differences for Shannon index comparing Winnie to APC^Min/+^ and Winnie-APC^Min/+^ samples (*p* = 0.0286) as well as in OTU (operational taxonomic unit) number comparing Winnie to Winnie-APC^Min/+^ mice (*p* = 0.0167) (Appendix A
Appendix A). At the phylum level (Figure 1), Winnie-APC^Min/+^ mice (Wi_APCMin) reported high abundances of *Verrucomicrobia*. Winnie mice (Wi) did not show the same significant association in Verrucomicrobia abundance, whereas both Winnie-APC^Min/+^ and Winnie reported a significant reduction in Deferribacteres and Proteobacteria abundances than wild type (WT) and APC^Min/+^ mice (APCMin) (*p* ≤ 0.002; *q* ≤ 0.012; Table 1). Concerning Firmicutes to Bacteroidetes ratios, we observed a significant reduction in Winnie-APC^Min/+^ mice compared to WT mice (*p* = 0.0181, Appendix A). Bacterial families that mainly characterized Winnie-APC^Min/+^ mice were Acholeplasmataceae, Bacteroidaceae, some members of Clostridiaceae (specifically clostridia not belonging to the Clostridium cluster I), Eubacteriaceae, Prevotellaceae, and Verrucomicrobiaceae (*p* ≤ 0.005; *q* ≤ 0.019; Table 1 and Appendix A). Conversely, Winnie-APC^Min/+^ mice reported a significant negative association with abundances of Clostridiaceae (cluster I), Coriobacteriaceae, Deferribacteraceae, Helicobacteraceae, and Lactobacillaceae (*p* ≤ 0.005; *q* ≤ 0.021; Table 1). Similarly to Winnie-APC^Min/+^, Winnie mice reported a positive association with Acholeplasmataceae, Clostridiaceae (not belonging to the Clostridium cluster I), and Eubacteriaceae abundances (*p* ≤ 0.001; *q* ≤ 0.004) and a negative association with Deferribacteraceae and Helicobacteraceae (*p* ≤ 0.001; *q* ≤ 0.004; Table 1). Only Winnie mice reported a negative association to Enterobacteriaceae abundances (*p* = 0.004; *q* = 0.017; Table 1).

As observed in Table 1, the principal component analysis (PCA), performed using as variables all bacterial families with a relative abundance > 1% at least in one sample, also showed a partial overlapping of Winnie-APC^Min/+^ and Winnie gut microbiota (Figure 2). Differences occurring between these two groups were related to *Bacteroidaceae*, *Prevotellaceae*, and *Verrucomicrobiaceae* abundances, mainly characterizing Winnie-APC^Min/+^ rather than Winnie mice (Figure 2; Table 1). The negative association with *Coriobacteriaceae* and *Lactobacillaceae* was significantly found only in Winnie-APC^Min/+^ mice (Figure 2; Table 1). Meanwhile, the afore reported negative association with *Proteobacteria* was specifically related to *Clostridiaceae* (cluster I) abundances in Winnie-APC^Min/+^ mice and to *Enterobacteriaceae* abundances in Winnie mice (Figure 2; Table 1).

In order to identify genus clusters characterizing gut microbiota of Winnie-APC^Min/+^, Winnie, APC^Min/+^, and C57BL/6J (WT) mice, we performed a permutation analysis (Figure 3). Based on the clustering of bacterial genera into three clusters (A, B, and C), mice were clustered in two different clusters (1 and 2). One APC^Min/+^ sample (APC^Min^52) was purged out as outlier based on genera included into cluster “C”. Therefore, the cluster 1 included only Winnie-APC^Min/+^ and Winnie mice, which showed high scores of genera belonged to cluster “A” and negative scores of genera belonged to cluster “B”. Conversely, cluster 2 included the other three APC^Min/+^ samples and all the four WT mice. This resulted by the high abundances of genera included into cluster “B” (in particular the subcluster “B2”), related to negative scores of genera included into cluster “A”.

As reported in Table 1 and Appendix A, both Winnie-APC^Min/+^ and Winnie mice showed a significantly positive association (*p* ≤ 0.001; *q* < 0.02) with *Acholeplasma*, *Alkaliphilus*, *Alloprevotella*, *Eubacterium*, *Paraprevotella*, and *Porphyromonas* (all included in the cluster “A”, except *Eubacterium*). Additionally, both shared the negative association with *Helicobacter*, *Odoribacter*, *Mucispirillum*, and *Ureaplasma* abundances (*p* ≤ 0.006; *q* ≤ 0.032), included into subcluster “B2”, except *Ureaplasma*. Meanwhile, only Winnie-APC^Min/+^ reported a positive association with *Akkermansia*, *Bacteroides*, and *Prevotella* (*p* ≤ 0.005; *q* ≤ 0.027) and negative with *Clostridium sensu stricto*, lactobacilli, *Rikenella*, *Stomatobaculum* and *Syntrophococcus* abundances (*p* ≤ 0.008; *q* ≤ 0.042). Differently, only Winnie mice reported a positive association with *Porphyromonas* (*p* = 0.001; *q* = 0.012) and negative with the *Escherichia/Shigella* pattern (*p* = 0.008; *q* = 0.042).

Based on obtained results, we noted that all significant associations (*p*- and *q*-values < 0.05) were related to Winnie and/or Winnie-APC^Min/+^ mice. Thus, in order to ascertain which OTUs mainly characterized microbiota of Winnie and/or Winnie-APC^Min/+^, we performed a statistical analysis also at the species level, investigating OTUs with a mean value of relative abundance > 1%. As reported in Table 2, we found 10 OTUs that significantly colonized the microbiota of Winnie, Winnie-APC^Min/+^ mice, or both. A total of 5 out of these 10 OTUs showed the same trend in both Winnie and Winnie-APC^Min/+^ mice. In detail, *Eubacterium coprostanoligenes*, *Paraprevotella clara*, and *Prevotellamassilia timonensis* were positively associated (*p* < 0.000; *q* ≤ 0.014), whereas *Lactobacillus intestinalis* and *Mucispirillum schaedleri* were negatively associated (*p* < 0.000; *q* ≤ 0.014; Table 2). *Ruminococcus champanellensis* was found positively associated only to Winnie mice (*p* = 0.001; *q* = 0.034). In Winnie-APC^Min/+^ mice, *Akkermansia muciniphila* and *Prevotella oralis* were positively associated (*p* ≤ 0.001; *q* ≤ 0.03), whereas *Clostridium leptum* and *Rikenella microfusus* were negatively associated (*p* < 0.001; *q* ≤ 0.036).

### 2.2. The Genetics of Breeding Couple Shapes APC^Min/+^ Mice Intestinal Microbiota

Even if APC^Min/+^ mice mainly developed small intestinal tumors and had a low rate of tumor incidence in the large intestine (as also demonstrated for mice housed in our animal facility) [43,44], we noticed an increased incidence of ACF in the distal colon of APC^Min/+^ mice born from heterozygous breeders for the Winnie mutation [40]. Thus, we decided to investigate the role of breeding couple genetics in ACF development. We firstly performed a histological analysis on the large intestine of 8-week-old APC^Min/+^ mice born from WT female and APC^Min/+^ male breeders. Table 3 shows the absence of dysplastic ACF incidence in the distal colon of APC^Min/+^ mice born from WT breeders as compared to those born from heterozygous breeders for the Winnie mutation [40]. However, occasional non dysplastic and dysplastic ACF in the form of unicryptic lesions and microadenoma > 5 crypts low-grade (LG) were detected in the medial colon of APC^Min/+^ mice born from WT breeders (Table 3).

Then, we investigated if the increase in dysplastic ACF incidence in APC^Min/+^ mice born from heterozygous breeders for Winnie mutation could be triggered by a “dysbiotic state” of the breeders. In fact, as we recently demonstrated, an intestinal dysbiosis has been observed in the offspring of heterozygous breeders for Winnie mutation [42]. Thus, in order to evaluate how maternal genotype and the relative gut microbiota could impact the gut microbial composition and the development of ACF in APC^Min/+^ mice, feces of APC^Min/+^ mice born from Winnie^+/−^ (*n* = 4) or WT (*n* = 2) mothers was collected at 4 and 8 weeks after birth. With the aim to identify bacterial taxa colonizing gut microbiota of the above reported APC^Min/+^ mice, we performed a clustering analysis using as variables bacterial genera with a relative abundance (16S rDNA gene amplicon) > 1% at least in one fecal sample. The permutation analysis (Figure 4) showed as both sampled time (4 and 8 week after birth) of APC^Min/+^ mice born from WT mothers clustered together. Moreover, only one APC^Min/+^ mouse born from Winnie^+/−^ was included in this cluster; all the other APC^Min/+^ mice born from Winnie^+/-^ clustered apart. The main differences between these two clusters were related to three bacterial clusters (I, II, and III; Figure 4). In detail, the bacterial cluster I, including *Helicobacter* and taxa belonging to the Clostridium cluster XIVa, mainly characterized APC^Min/+^ mice with WT parental backgrounds. The same was found also concerning the subcluster IIIa, which included Prevotella, Lactobacillus, and Eisenbergiella. On the other side, the cluster II, which included Bacteroides and Odoribacter, was mainly related to APC^Min/+^ mice born from Winnie^+/−^ mothers. Of note, within the subcluster IIIb, which included bacterial genera not absolutely correlated to one mouse cluster, Parabacteroides and Akkermansia seem to show a trend towards the relationship with APC^Min/+^ mice born from Winnie^+/−^ mothers.

### 2.3. Fecal Transplant from Winnie to APC^Min/+^ Mice Triggers ACF Development

To test the hypothesis of dysbiotic microbiota as a transmittable factor for ACF development, we performed a fecal transplant from Winnie donors into APC^Min/+^ mice born from WT mothers. APC^Min/+^ mice were treated with broad spectrum antibiotics for two weeks and gavaged with a suspension of fecal material from WT or Winnie donors for the following two weeks. At the end of the experiment, mice were sacrificed, and the colon explanted and histologically analyzed to evaluate ACF development (Figure 5).

Compared to untreated APC^Min/+^ mice, fecal transplant of Winnie microbiota induced the formation of large tumor lesions in the distal colon leading to partial intestinal obstruction (Figure 5B, right panel and F), and a tendency to splenomegaly (Figure 5C). Colons from APC^Min/+^ mice after fecal transplant from Winnie also showed watery stool, as compared to control groups (Figure 5B, left panel). Table 4 showed the highest incidence of dysplastic ACF, ranging from unicryptic lesion to microadenoma >5 HG, in the distal colon of APC^Min/+^ mice after fecal transplant from Winnie mice. However, the transplant of fecal homogenate from Winnie mice also induced the onset of non dysplastic ACF as well as dysplastic microadenoma >1 ≤5 LG and microadenoma >5 LG in the medial colon of APC^Min/+^ mice. Moreover, non dysplastic ACF were also seen in the medial colon of APC^Min/+^ mice treated with fecal transplant from WT donors, whilst no dysplastic lesions were observed in this experimental group.

## 3. Discussion

Numerous reports highlight the link between intestinal microbiota and CRC [45]. Based on our results, the Winnie-APC^Min/+^ model is characterized by an intestinal microbial composition that mainly resembles characteristics of the parental strain Winnie. This was particularly shown by the β-diversity analysis, where we observed the clear microbial proximity between Winnie and Winnie-APC^Min/+^. This finding also supports what we previously observed in terms of body weight for Winnie-APC^Min/+^ and Winnie mice [42].

Our breeding strategy relies on heterozygote mothers for Muc2 point mutation (Winnie^+/−^ mice) characterized by a mild dysbiosis transmittable to the offspring and the acquisition of distinct characteristics dictated their own genetics at 8 weeks. One of the main features characterizing Winnie mice is the increased abundance of *Akkermansia muciniphila* [42]. Additionally, Winnie mice have a less-tight mucus layer providing an easier access to mucin metabolized by mucin degrader bacteria, such as *A. muciniphila*, as a source of energy [46]. Of note, in the present work *Akkermansia* was associated with Winnie-APC^Min/+^ even more significantly than Winnie mice. This might be interesting, considering that in dextran sodium sulfate (DSS)-induced colitis, *Akkermansia* showed the capability to produce extracellular vesicles protecting the host from colitis progression [47]. However, we want to emphasize that further studies are needed to deeply understand how *Akkermansia* contributes (protecting or promoting) to a specific subset of pathologies, particularly in chronic intestinal inflammations and related ones.

Apart from *Akkermansia*, data obtained from gut microbiota sequencing (16S rDNA gene amplicon) indicated that Winnie-APC^Min/+^ mice harbor a bacterial pattern mainly characterized by *Bacteroidaceae*, *Prevotellaceae, Acholeplasmataceae*, *Eubacteriaceae*, and clostridia not belonging to *Clostridium sensu stricto*. Conversely, a low abundance of butyrogenic bacteria was found, specifically *Bifidobacteriaceae*, *Lactobacillaceae*, *Lachnospiraceae*, *Ruminococcaceae,* and the relative subtaxa. These findings were shared with Winnie mice, whereas APC^Min/+^ and C57BL/6J showed exactly the opposite. The increase in *Bacteroides* and *Prevotella* abundances in Winnie-APC^Min/+^ compared to WT mice strengthens the power of this model in resembling the human pathology, considering the evidence found in CRC patients when compared to healthy controls [48]. Moreover, gut microbiota sequencing of Winnie-APC^Min/+^ mice confirmed an early important feature of the intestinal dysbiosis leading to CRC in humans, i.e., the decrease in *Firmicutes* to *Bacteroidetes* ratio relative to WT mice, in accordance with data found in subjects with preneoplastic/neoplastic lesions [49]. Under another point of view, *Bacteroides* have been previously correlated with a low hypermethylation of several gene promoters, including *Wif1* (Wnt Inhibitory Factor 1) for the Wnt pathway and Neuropeptide Y (NPY) and Proenkephalin (PENK) for the brain gut system, as recently demonstrated in a large cohort of patients [50]. In the same work, the aforementioned butyrogenic bacteria were associated with a higher cumulative methylation index (CMI) indicating gene expression silencing, by gene methylation, relative to the host genes involved in CRC pathways [50]. Taking into account these recent findings and, in general, the role of epigenetics in CRC onset and progression [51,52], it is tempting to speculate that the dysbiosis of Winnie-APC^Min/+^ mice also supported ACF development via an epigenome dysregulation.

The protective role of SCFA-producing bacteria is widely recognized [53,54]. Among SCFAs, butyrate exerts a beneficial effect on intestinal inflammation due to its effect on tight junctions in maintaining the integrity of the epithelial barrier [55,56] and therefore favoring mucus excretion by enterocytes. Even if the protective role of butyrate-producing bacteria could be controversial in some cases [57], in our study the beneficial role also emerges by comparing the microbial profiling of APC^Min/+^ mice born from WT or Winnie^+/−^ breeders. In this line, maternal dysbiosis in Winnie mice could be considered as a trigger acting on the genetic background of APC^Min/+^ mice to promote ACF development. The loose mucus layer, typical of Winnie mice, increases the intestinal permeability allowing bacteria to invade the mucosa with a consequent activation of the immune response. Thus, all these microenvironmental modulations lay the ground for specific bacteria (e.g., *Bacteroides*) to act as inflammatory and tumorigenic triggers. Different *Bacteroides* taxa are, indeed, lipopolysaccharide (LPS) producers [58]. Moreover, LPS has been found to induce IL-8 activation in human intestinal epithelial cells and this has been related to specific histone acetylation and methylation changes [59]. Therefore, this evidence, combined with the low grade of SCFA excretion and the derived easy contact of microbial antigens to the intestinal epithelium, may boost inflammatory features as well as the formation of ACF. Moreover, based on the protective role of *Mucispirillum schaedleri* in different mice diseases [60], the lowest amount of this bacterial strain in Winnie and Winnie-APC^Min/+^ models may contribute to CAC and thus warrants further investigation. However, due to the lack of genome and transcriptome analyses, we would underline that the pathogenicity is not related only to the presence/absence of pathobiont, more than probiotic, taxa. Meanwhile, we would emphasize that this is the result of a multifactorial interaction, in which the host genotype, combined with the lack of butyrogenic bacteria and the related expansion of other taxa replacing beneficial microbes, drives the onset of CRC.

Even if our data are in line with the dysbiotic pattern supporting CRC pathogenesis, we acknowledge some discrepancies relative to the recent literature. In fact, differently from what has been reported in other studies [61,62], in our experimental setup the *Erysipelotrichaceae* was not among the most represented family and we even detected a slight increase in the Winnie-APC^Min/+^ model. These data suggest that variability among the animal facility represents a crucial difference that will require further investigation. On the other hand, different models may lead to different results representing distinct but relevant risk factors for CRC development.

Our current data are still not sufficient to provide the final proof that dysbiosis-mediated inflammation is the main trigger of CAC development in Winnie-APC^Min/+^ mice, although timing and location of the ACF represent a strong hint supporting this hypothesis. Furthermore, the observed results should be discussed considering that the ACF with low incidence observed in the APC^Min/+^ offspring of Winnie^+/−^ mothers may be the result of a cascade of events dictated by the presence/absence of specific intestinal taxa that leads to inflammation and tumor lesion development. In fact, in the distal colon of 8-week-old APC^Min/+^ offspring from WT mothers, ACF are extremely rare [43]. Considering this evidence, it is tempting to speculate that microbiota supports CRC onset and, therefore, may represent a risk factor for transmittable susceptibility to ACF development in genetically predisposed individuals.

We would also support the idea that Winnie-APC^Min/+^ mice microbiota profiling could help to define adjuvant treatments targeting gut microbiota imbalances. These treatments, combined with common chemotherapy and/or immunotherapy strategies, might ameliorate CRC treatment efficacy. Additionally, probiotics and prebiotics, defined as live microorganisms promoting beneficial effects to the host and indigestible fibers harboring the growth of health-promoting bacteria (respectively), may drive or reduce an overexpansion of undesirable microbes. Both these non-medical therapies could contribute, as adjuvant treatments, to regulating the immune response, macrophage and lymphocyte activation, and the proliferation/differentiation of intestinal epithelial cells [35]. The end goal remains undoubtedly a beneficial influence towards the intestinal barrier function. Furthermore, diet is recognized as one of the most influencing factors able to shape gut microbiota with a health-promoting effect associated with a high intake of food enriched in fiber and bioactive compounds, e.g., polyphenols, characterized by a well-known anti-inflammatory effect [63].

Taken together, the data reported in the present study could be useful for the design of appropriate microbiota-based screening and preventive strategies for genetically predisposed individuals in order to avoid CRC onset and/or progression. Together with others, we recently demonstrated that nutritional-based strategies could be effective in bending intestinal microbiota towards an eubiotic state (balanced microbial composition). This possibility should be explored as a preventive/adjuvant strategy [63,64,65]. Positive results may also help to reduce the huge financial costs for the health system related to ever increased number of diagnoses for inflammatory diseases as well as for their related pathological conditions.

It is important to note that our results referred to fecal specimens. Even if the analysis of tissue samples from colonic mucosa seems to be more valuable to study CRC physiopathology, many of the studies on gut microbiota are performed on fecal specimens due to the non-invasive nature of sample collection.

Finally, further studies on microbiota modulation in the Winnie-APC^Min/+^ model have to be extended to the non-bacterial components of the microbiome (i.e., fungi, protozoans, viruses). In fact, even if the pathogenetic role for CRC of these microorganisms is already known, the knowledge of their mechanisms of action is still underestimated probably due to their lower abundances and/or the technical difficulties in targeting them [35]. Future perspectives for this study will also include the use of complementary approaches, such as metatranscriptomics, metabolomics and metaproteomics, to provide a deeper comprehension of host–microbes interactions. Currently, a multiomics approach is ongoing to evaluate the crosstalk between intestinal resident microbiota and the intestinal epithelium, with a particular interest in the metabolites produced in close proximity to the tumor lesions.

## 4. Materials and Methods

### 4.1. Mice

Animal studies were conducted in accordance with national and international guidelines and were approved by the authors’ institutional review board (Organism For Animal Wellbeing—OPBA). All animal experiments were carried out in accordance with Directive 86/609 EEC enforced by Italian D.L. n. 26/2014, and approved by the Italian Animal Ethics Committee of Ministry of Health—General Directorate of Animal Health and Veterinary Drugs (DGSAF- Prot. 768/2015-PR 27/07/2015). Animals were sacrificed if found in severe clinical condition to avoid undue suffering.

The new murine transgenic line Winnie-APC^Min/+^ was created by breeding Winnie^+/−^ mice with APC^Min/+^ mice on a C57BL/6J background. WT and APC^Min/+^ mice murine lines were purchased from Jackson Laboratories (C57BL/6J, Stock No. 000664, C57BL/6J-APCMin/J, Stock No. 002020, respectively) (Bar Harbor, ME, USA). Winnie mice were obtained from the University of Tasmania, Launceston, Australia (Dr R. Eri’s laboratory). The 4 experimental groups are composed of 4 mice for the C57BL/6J (wild type, WT), Winnie, and APC^Min/+^ groups and 7 mice for the Winnie-APC^Min/+^ group.

Mice were sacrificed at 8 weeks of age; colons were removed to evaluate the presence of neoplasia and metagenetic analyses was performed on stool.

### 4.2. Histology

Tissue sections from the large intestine were fixed in 10% buffered formalin, dehydrated, and paraffin embedded. Then, 3-µm-thick sections from proximal, medial, and distal colon were stained using a hematoxylin/eosin standard protocol. Colonic tissue sections were evaluated for neoplasia. Observations and imaging were performed with a Nikon Eclipse Ti2.

### 4.3. DNA Extraction from Stool

Total genomic bacterial DNA was isolated from frozen stool samples of 8-week-old mice using the QIAamp^®^Fast DNA Stool Mini Kit (QIAGEN, Hilden, Germany), according to the manufacturer’s instructions.

### 4.4. S rDNA Metagenomic Library Preparation, Sequencing and Analysis

Next generation sequencing experiments, comprising DNA extraction and primary bioinformatics analysis, were performed by Genomix4life S.R.L. (Baronissi, Salerno, Italy). DNA extractions were performed with Invimag Stool kit (Stratec, Birkenfeld, Germany), using an extraction negative control. Final yield and quality of extracted DNA were determined by using NanoDrop ND-1000 spectrophotometer (Thermo Scientific, Waltham, MA, USA) and Qubit Fluorometer 1.0 (Invitrogen Co., Carlsbad, CA, USA). 16S amplification was performed with primers: Forward: 5′-CCTACGGGNGGCWGCAG-3′ and Reverse: 5′-GACTACHVGGGTATCTAATCC-3′ [66], which target the hypervariable V3 and V4 region of the 16S rRNA gene. Each PCR reaction was assembled according to Metagenomic Sequencing Library Preparation (Illumina, San Diego, CA, USA). A negative control is included in the workflow; it consists of all reagents used during sample processing (16S amplification and library preparation) but does not contain a sample, to ensure no contamination. Libraries were quantified used Qubit fluorometer (Invitrogen Co., Carlsbad, CA, USA) and pooled to an equimolar amount of each index-tagged sample to a final concentration of 4 nM, including the Phix Control Library. Pooled samples were subject to cluster generation and sequenced on MiSeq platform (Illumina, San Diego, CA, USA) in a 2 × 300 paired-end format. The raw sequence files generated (fast files) underwent quality control analysis with FastQC.

The 16S rDNA analysis performs taxonomic classification of 16S rRNA targeted amplicon reads. The algorithm is a high-performance implementation of the Ribosomal Database Project (RDP) Classifier described in [67]. Taxonomic databases to perform taxonomic classification, after OTU clustering with a 97% of coverage (3% of divergence), is RefSeq RDP 16S v3 May 2018 DADA2 32bp.

### 4.5. Fecal Transplantation

APC^Min/+^ mice (*n* = 10) were treated with broad spectrum antibiotics in their drinking water, which was replaced daily for 2 weeks. After the antibiotic treatment, mice returned to drink fresh water and were gavaged with 0.2 mL of supernatant from a fresh fecal homogenate pool of 8-month-old WT or Winnie donor mice (3 or 7 APC^Min/+^ mice were treated, respectively; n of donor mice = 4 for each group). Fecal transplant was performed every 2 days for 2 weeks with a stainless-steel tube without prior sedation of the mice. Mice were sacrificed at the end of the experiment and the colon explanted for histological analysis.

### 4.6. Statistical Analysis

Statistical analysis of fecal transplant experimental data was performed using the Graphpad Prism statistical software release 5.0. All data were expressed as means ± S.E.M. Statistical significance was evaluated with two-tailed Student’s t-test and results were considered statistically significant at *p* < 0.05.

For microbiota analysis, data were summarized using descriptive statistics, such as means and standard deviations, median, or interquartile range (IQR), as appropriate, for quantitative variables and relative frequencies for qualitative ones. Multivariable association between 16S rDNA gene data abundances at different taxonomic levels occurring in mice microbiota was performed using the MaAsLin2 R package (https://huttenhower.sph.harvard.edu/maaslin/). The *Firmicutes* to *Bacteroidetes* ratios was analyzed by Kruskal–Wallis test corrected with Dunn’s multiple comparisons test with a significance level of *p* < 0.05. Meanwhile, unless specifically described, data and group differences were analyzed by paired or unpaired, two-tailed Student’s *t*-test. Principal component analysis (PCA) [68,69] was carried out using the statistical software Statistica for Windows (Statistica 6.0 for Windows 1998, StatSoft, Vigonza, Italy) based on weighted Unifrac distances. Samples more similar to each other should appear closer together according to the respective axis reflecting the variation among all samples [70]. This technique is useful for displaying clusters existing within data. The variables (features) reflect the relative bacterial composition in a sample at a particular taxonomic level. In addition, Permut-MatrixEN software was used to identify clusters at the level of the mouse groups and taxa [71]. Statistical analysis of the relative abundances of microbial genera was based on Duncan’s multiple range test, with a significance level of *p* < 0.05.

## 5. Conclusions

Our results clearly indicate that microbial components of the intestinal tract represent a sufficient trigger for CRC development in genetically predisposed individuals. The paradigm of CRC as non-communicable disease may change in light of these data, particularly when translated to the mother–child microbial transfer. Cancer prevention may soon include prevention of intestinal dysbiosis and treatment for “healthy microbial pattern” selection. We are just starting to understand the complex microbial–host interaction that is much more than just bacterial species-relative abundance. Our data will contribute to shedding light on the importance of microbiome modulation as a prevention strategy in CRC and, possibly, numerous other pathologies.

## Figures and Tables

**Figure 1 cancers-13-00283-f001:**
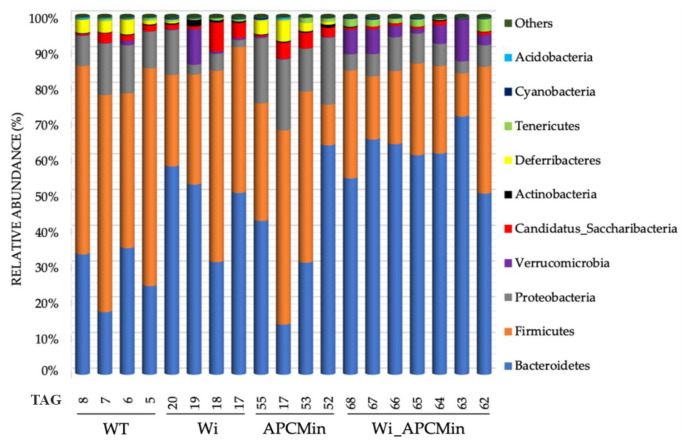
Gut microbiota profiling at phylum level of 8-week-old Winnie-APC^Min/+^ mice (Wi_APCMin) and their control littermates. Ear TAGs of each mouse for all the experimental groups are indicated. Abbreviations: WT, wild type—C57BL/6J mice; Wi, Winnie mice; APCMin, APC^Min/+^ mice.

**Figure 2 cancers-13-00283-f002:**
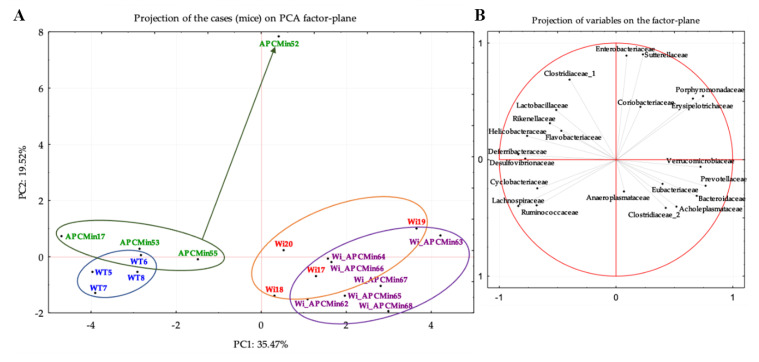
Principal component analysis (PCA). The left panel (**A**) shows β-diversity at the family level (16S rDNA gene amplicon) in 8-week-old Winnie-APC^Min/+^ (Wi_APCMin), Winnie (Wi), APC^Min/+^ (APCMin), and C57BL/6J mice (WT). The right panel (**B**) shows variables weight on PCA system based on normalized bacterial family abundances. Variables placed in both the right quadrants of the PCA system show positive standardized values of the first principal component (PC1). Variables showing negative standardized PC1 values were placed into the two left quadrants. Variables placed in both the upper quadrants of the PCA system show positive standardized values of the second principal component (PC2). Variables showing negative standardized PC2 values were placed into the two lower quadrants.

**Figure 3 cancers-13-00283-f003:**
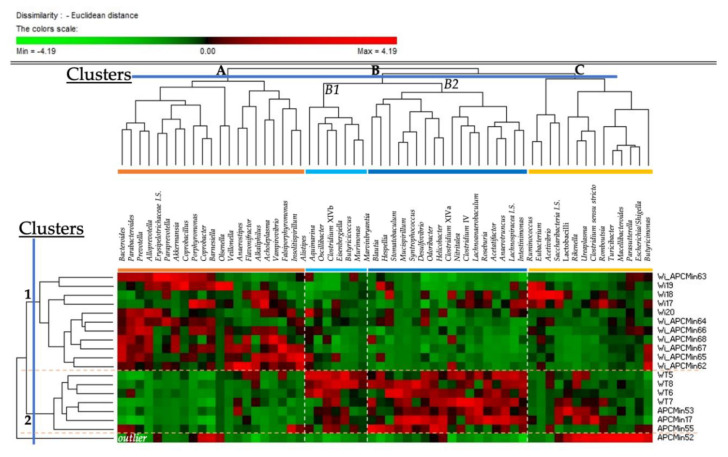
Permutation analysis of bacterial genera with a relative abundance (16S rDNA gene amplicon) > 0.5% in at least one 8-week-old Winnie-APC^Min/+^ (Wi_APCMin), Winnie (Wi), APC^Min/+^ (APCMin), or C57BL/6J (WT) mouse.

**Figure 4 cancers-13-00283-f004:**
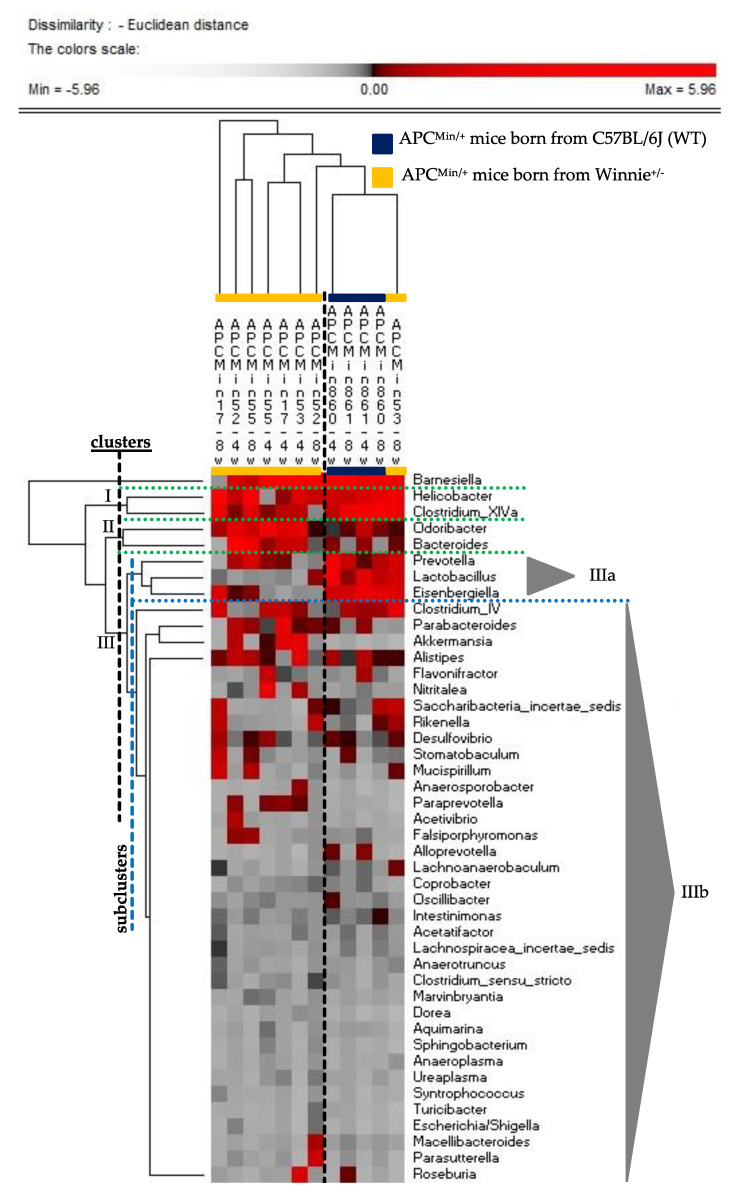
Permutation analysis of bacterial genera with a relative abundance (16S rDNA gene amplicon) >1% in at least one APC^Min/+^ mouse. Feces was sampled from APC^Min/+^ born from Winnie^+/−^ (*n* = 4; yellow) or C57BL/6J (WT; *n* = 2; blue) mice at 4 and 8 weeks after birth.

**Figure 5 cancers-13-00283-f005:**
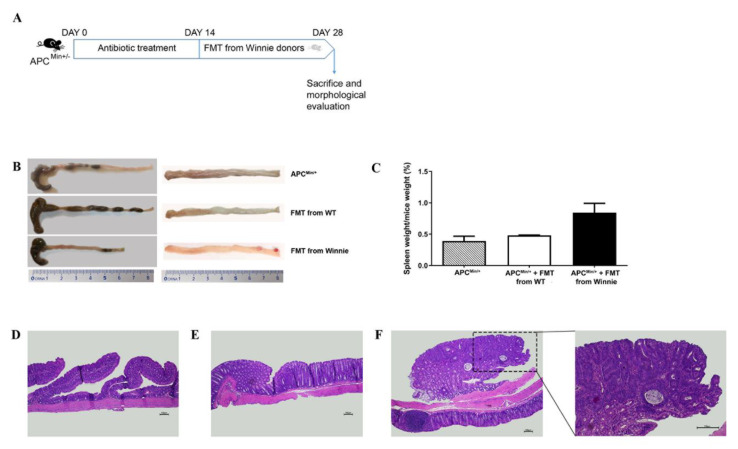
Fecal microbiota transplantation (FMT) from WT or Winnie donors in APC^Min/+^ mice born from WT mothers. (**A**) Experimental scheme of FMT protocol. (**B**) Representative image of colon explanted from untreated APC^Min/+^ and APC^Min/+^ mice after FMT treatment. (**C**) Ratio of spleen weight vs. mice weight was calculated for the three experimental groups. **(D**–**F**) Representative H&E staining on colon sections from proximal (**D**), medial (**E**), and distal (**F**) tract of APC^Min/+^ mice with FMT from Winnie mice. Images were captured at 10X and 20X magnifications. Scale bar = 100 μM.

**Table 1 cancers-13-00283-t001:** Significant differences (*p*-value < 0.05 and *q*-value < 0.05) of bacterial abundances (16 rDNA gene amplicon) among all 8-week-old sampled mice: Winnie-APC^Min/+^ (Wi_APCMin), Winnie (Wi), APC^Min/+^ (APCMin), or C57BL/6J (WT). The column “coef” shows positive (coef+) or negative (coef−) associations of bacterial taxa reported in the same row related to the different groups of mice (column “Mice genotype”). Interquartile ranges (IQR, 25th–75th percentile) of bacterial abundances in all analyzed groups are also reported (*n* = 4–7 animals/group).

	Mice Genotype	Taxon	coef	pval	qval	IQR Wi_APCMin	IQR Wi	IQR APCMin	IQR WT
Phylum	Wi_APCMin	Deferribacteres	−	0.0003	0.0031	0.06–0.13	0.01–0.13	1.94–4.34	2.67–3.81
		Proteobacteria	−	0.0022	0.0123	5.3–7.34	2.58–6.69	16.73–19.03	9.84–13.72
		Verrucomicrobia	+	0.0004	0.0035	3.06–6.9	0.46–2.94	0.11–0.23	0.07–0.42
	Winnie	Deferribacteres	-	0.0002	0.0031	0.06–0.13	0.01–0.13	1.94–4.34	2.67–3.81
		Proteobacteria	−	0.0006	0.0037	5.3–7.34	2.58–6.69	16.73–19.03	9.84–13.72
Family	Wi_APCMin	Acholeplasmataceae	+	0.0002	0.0034	0.37–1.12	0.36–0.57	0–0.01	0.01–0.04
		Bacteroidaceae	+	0.0045	0.0187	8.13–15.14	3.35–5.61	1.26–4.49	1.01–3.56
		Clostridiaceae_1	−	0.0006	0.0045	0.11–0.22	0.35–0.58	0.51–1.83	0.24–0.46
		Clostridiaceae_2	+	0.0027	0.0134	0.93–2.08	1.63–3.35	0.18–0.24	0.27–0.89
		Coriobacteriaceae	−	0.0028	0.0134	0.06–0.12	0.37–0.73	0.25–0.45	0.12–0.16
		Deferribacteraceae	−	0.0003	0.0034	0.06–0.13	0.01–0.13	2.08–4.57	2.83–3.98
		Eubacteriaceae	+	<0.0001	0.0031	0.61–1.16	2.48–4.23	0.03–0.07	0.08–0.09
		Helicobacteraceae	−	0.0053	0.0205	0.92–2.89	0.43–2.47	9.17–15.1	5.31–9.97
		Lactobacillaceae	−	0.0002	0.0034	0.28–0.56	0.87–1.47	1.46–4.42	1.46–2.04
		Prevotellaceae	+	0.0017	0.0100	12.95–20.68	9.23–12.92	1.02–6.65	1.99–4.92
		Verrucomicrobiaceae	+	0.0008	0.0054	3.17–7.15	0.49–3.11	0.07–0.21	0.01–0.36
	Winnie	Acholeplasmataceae	+	0.0005	0.0042	0.37–1.12	0.36–0.57	0–0.01	0.01–0.04
		Clostridiaceae_2	+	0.0005	0.0043	0.93–2.08	1.63–3.35	0.18–0.24	0.27–0.89
		Deferribacteraceae	−	0.0003	0.0034	0.06–0.13	0.01–0.13	2.08–4.57	2.83–3.98
		Enterobacteriaceae	−	0.0039	0.0170	0.02–0.17	0.01–0.01	0.09–0.49	0.01–0.04
		Eubacteriaceae	+	<0.0001	0.0003	0.61–1.16	2.48–4.23	0.03–0.07	0.08–0.09
		Helicobacteraceae	−	0.0018	0.0100	0.92–2.89	0.43–2.47	9.17–15.1	5.31–9.97
Genus	Wi_APCMin	Acholeplasma	+	0.0002	0.0037	0.39–1.19	0.38–0.58	0–0.01	0.01–0.04
		Akkermansia	+	0.0007	0.0066	3.36–7.52	0.52–3.29	0.06–0.22	0.01–0.38
		Alkaliphilus	+	0.0016	0.0124	0.97–2.18	1.71–3.33	0.1–0.23	0.25–0.89
		Alloprevotella	+	<0.0001	<0.0001	1.59–2.55	0.98–2.52	0.01–0.1	0–0.01
		Bacteroides	+	0.0045	0.0267	8.63–15.00	3.52–6.01	1.33–4.66	1.07–3.79
		Clostridium sensu stricto	−	0.0007	0.0066	0.11–0.21	0.34–0.54	0.51–1.85	0.23–0.48
		Eubacterium	+	<0.0001	0.0008	0.62–1.18	2.6–4.28	0.03–0.05	0.05–0.07
		Helicobacter	−	0.0059	0.0323	0.98–3.05	0.41–2.62	9.41–15.65	5.67–10.43
		Lactobacillus	−	0.0002	0.0037	0.3–0.59	0.92–1.51	1.47–4.62	1.55–2.12
		Mucispirillum	−	0.0003	0.0041	0.06–0.14	0.01–0.14	2.11–4.71	2.92–4.24
		Odoribacter	−	0.0023	0.0155	0.99–1.42	0.71–1.6	2.22–4.5	3.26–3.65
		Paraprevotella	+	<0.0001	0.0012	0.80–2.87	1.07–2.15	0.01–0.06	0.02–0.09
		Porphyromonas	+	0.0021	0.0150	0.41–0.64	0.39–0.92	0.09–0.2	0.05–0.09
		Prevotella	+	0.0038	0.0242	9.11–15.07	6.07–8.24	0.62–6.54	1.96–5
		Rikenella	−	0.0002	0.0037	0.06–0.45	0.49–1.01	2.92–5.48	1.53–3.8
		Stomatobaculum	−	0.0009	0.0077	0.08–0.15	0.18–0.57	1.28–3.33	2.18–5.3
		Syntrophococcus	−	0.0080	0.0418	0.01–0.06	0.02–0.05	0.2–0.59	0.13–0.69
		Ureaplasma	−	0.0006	0.0066	0–0.04	0.01–0.06	0.28–0.57	0.14–0.25
	Winnie	Acholeplasma	+	0.0005	0.0053	0.39–1.19	0.38–0.58	0–0.01	0.01–0.04
		Alkaliphilus	+	0.0003	0.0038	0.97–2.18	1.71–3.33	0.1–0.23	0.25–0.89
		Alloprevotella	+	<0.0001	0.0003	1.59–2.55	0.98–2.52	0.01–0.1	0–0.01
		Escherichia/Shigella	−	0.0082	0.0418	0.01–0.17	n.d.*	0.09–0.49	0.01–0.04
		Eubacterium	+	<0.0001	0.0001	0.62–1.18	2.6–4.28	0.03–0.05	0.05–0.07
		Helicobacter	−	0.0019	0.0137	0.98–3.05	0.41–2.62	9.41–15.65	5.67–10.43
		Mucispirillum	−	0.0003	0.0038	0.06–0.14	0.01–0.14	2.11–4.71	2.92–4.24
		Odoribacter	−	0.0052	0.0293	0.99–1.42	0.71–1.6	2.22–4.5	3.26–3.65
		Paraprevotella	+	0.0002	0.0037	0.80–2.87	1.07–2.15	0.01–0.06	0.02–0.09
		Porphyromonas	+	0.0014	0.0115	0.41–0.64	0.39–0.92	0.09–0.2	0.05–0.09
		Ureaplasma	−	0.0040	0.0242	0–0.04	0.01–0.06	0.28–0.57	0.14–0.25

* n.d.: not detected.

**Table 2 cancers-13-00283-t002:** Positive (coef +) or negative (coef -) significant associations (*p*-value < 0.05 and *q*-value < 0.05) of bacterial species with a mean abundance > 1% (16rDNA gene amplicon) among all 8-week-old sampled mice: Winnie-APC^Min/+^ and Winnie compared to APC^Min/+^, and C57BL/6J (WT) (*n* = 4–7 animals/group).

OTU	Group	coef	pval	qval
Akkermansia muciniphila	Winnie-APC^Min/+^	+	0.0007	0.0303
Clostridium leptum (Clostridium cluster IV)	Winnie-APC^Min/+^	−	0.0015	0.0455
Eubacterium coprostanoligenes	Winnie-APC^Min/+^	+	<0.0001	0.0007
Lactobacillus intestinalis	Winnie-APC^Min/+^	−	<0.0001	0.0079
Mucispirillum schaedleri	Winnie-APC^Min/+^	−	0.0003	0.0142
Paraprevotella clara	Winnie-APC^Min/+^	+	<0.0001	0.0043
Prevotella oralis	Winnie-APC^Min/+^	+	<0.0001	0.0043
Prevotellamassilia timonensis	Winnie-APC^Min/+^	+	<0.0001	0.0006
Rikenella microfusus	Winnie-APC^Min/+^	−	0.0002	0.0116
Eubacterium coprostanoligenes	Winnie	+	<0.0001	0.0004
Lactobacillus intestinalis	Winnie	−	0.0002	0.0120
Mucispirillum schaedleri	Winnie	−	0.0002	0.0122
Paraprevotella clara	Winnie	+	0.0003	0.0142
Prevotellamassilia timonensis	Winnie	+	<0.0001	0.0033
Ruminococcus champanellensis	Winnie	+	0.0010	0.0336

**Table 3 cancers-13-00283-t003:** Histological analysis of proximal (PC), medial (MC), and distal (DC) colon from 8-week-old APC^Min/+^ mice born from WT mothers. The incidence and multiplicity ± SEM of the selected tissues were calculated for non dysplastic and dysplastic ACF. The score for dysplastic ACF was calculated relative to all groups and to each group of neoplastic lesions classified according to dimension and grading of unicryptic lesions: microadenoma >1 ≤5 crypts low-grade (LG); microadenoma >5 crypts low-grade (LG); and microadenoma >5 crypts high grade (HG).

Genotype-Tissue (n. Mice)	Non Dysplastic ACF		Dysplastic ACF									
			All Groups		Unicryptic Lesion		Microadenoma >1 ≤5—LG		Microadenoma >5—LG		Microadenoma >5—HG	
	Incidence (%)	Multiplicity (Mean ± SEM)	Incidence (%)	Multiplicity (Mean ± SEM)	Incidence (%)	Multiplicity (Mean ± SEM)	Incidence (%)	Multiplicity (Mean ± SEM)	Incidence (%)	Multiplicity (Mean ± SEM)	Incidence (%)	Multiplicity (Mean ± SEM)
APC^Min/+^—PC (2)	0	0	0	0	0	0	0	0	0	0	0	0
APC^Min/+^—MC (2)	50	0.5 ± 0.5	100	1.5 ± 0.5	50	0.5 ± 0.5	0	0	100	1.0 ± 0	0	0
APC^Min/+^—DC (2)	0	0	0	0	0	0	0	0	0	0	0	0

**Table 4 cancers-13-00283-t004:** Histological analysis of APC^Min/+^ mice after two weeks of fecal microbiota transplantation (FMT) from WT or Winnie donors. The incidence and multiplicity ± SEM of proximal (PC), medial (MC), and distal (DC) colon were calculated for non dysplastic and dysplastic ACF. The score for dysplastic ACF was calculated relative to all groups and to each group of neoplastic lesions classified according to dimension and grading of unicryptic lesions: microadenoma >1≤ 5 crypts low-grade (LG); microadenoma >5 crypts low-grade (LG); and microadenoma >5 crypts high grade (HG).

Genotype Tissue Treatment (n. Mice)	Non Dysplastic ACF		Dysplastic ACF									
			All Groups		Unicryptic Lesion		Microadenoma >1 ≤5 —LG		Microadenoma >5—LG		Microadenoma >5—HG	
	Incidence (%)	Multiplicity (Mean ± SEM)	Incidence (%)	Multiplicity (Mean ± SEM)	Incidence (%)	Multiplicity (Mean ± SEM)	Incidence (%)	Multiplicity (Mean ± SEM)	Incidence (%)	Multiplicity (Mean ± SEM)	Incidence (%)	Multiplicity (Mean ± SEM)
APC^Min/+^—PC with FMT from WT (3)	0	0	0	0	0	0	0	0	0	0	0	0
APC^Min/+^—PC with FMT from Winnie (7)	0	0	0	0	0	0	0	0	0	0	0	0
APC^Min/+^—MC with FMT from WT (3)	33.3	0.33 ± 0.33	0	0	0	0	0	0	0	0	0	0
APC^Min/+^—MC with FMT from Winnie (7)	14.3	0.14 ± 0.14	42.6	0.43 ± 0.2	0	0	28.6	0.29 ± 0.19	14.3	0.14 ± 0.14	0	0
APC^Min/+^—DC with FMT from WT (3)	0	0	0	0	0	0	0	0	0	0	0	0
APC^Min/+^—DC with FMT from Winnie (7)	14.3	0.86 ± 0.86	42.6	0.86 ± 0.6	14.3	0.14 ± 0.14	14.3	0.14 ± 0.14	28.6	0.14 ± 0.37	28.6	0.29 ± 0.19

## Data Availability

The data presented in this study are available on request from the corresponding author.

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
