# Peer review of "Dysbiosis Triggers ACF Development in Genetically Predisposed Subjects"

_cancers, 2021, doi:10.3390/cancers13020283_

Round 1

Reviewer 1 Report

Please apply bold letters to designate taxonomy orderly.

Maybe, you should use ASVs instead of OTUs.

Please define in the sample summary what does ACF (aberrant crypt foci) stand for.

In the Materials and Methods section maybe you should provide the trimming parameters and specify which reference database was used.

"28,301.18 ± 128 17,409.02 number of reads per samples, of which 89.09 ± 2.12 was assigned to genus level". Are the number of reads (89.09 ± 2.12) identified to the genus level relative ratios?

In Figure 1 define the meaning of numbers (8, 7, 6, 5 / 20, 19, 08, 17 etc.). Also, here, I do not see the “n” numbers you are referring to in the figure 1 description.

In Table 1, did you work with absolute or relative frequency data? Also, here I do not see the reason to investigate “others”. I found this table too complex, maybe you should upload it as a supplementary material and represent correlation coefficients in a heatmap.

In Figure 2. does your Principal component analysis based on weighted or unweighted unifrac distances? Also, I do not see the “n” numbers, but I think the number of animals in the experimental group should be mentioned in the Materials and Methods section only. Also, does Figure 2 contain an a) and a b) part? If so, part b) “projection of variable on the factor plane” is not readable and you should provide more information about it.

Figure 3 is very hard to read. Maybe, by rotating the names of the taxa by 90 degrees could help the readers’ eyes.

You wanted to prove a hypothesis that gut microbiota can be important transmittable factor in CRC. To prove this, you performed a fecal transplant from Winnie donors into Apc(Min/+) mice born from WT mothers. For these, before fecal transplantation you treated Apc(min/+) mice with broad spectrum antibiotics for two weeks. Nevertheless, it is also known, that the exposure of broad-spectrum antibiotics can be linked to colon cancer. Therefore, I missed to see data about colon microbiota content and representative H&E staining at the end of the two-week antibiotic exposure as a point of reference.

In my opinion a few data about the Firmicutes-to-Bacteroidetes ratios could add up to the value of this research.

In this study, you also referred to short-chain fatty acids (SCFAs) as the main fermentation products of intestinal microbiota that provide link between the gut microbiota and the host’s physiology. Why did not you report about SCFA contents in your samples? Do you have these data? I my view you should have been investigated a correlation among gut dysbiosis, SCFAs.

Reviewer 2 Report

I commend the authors. This is good science and it is interesting. I found the presentation and the processes sound.

I find the utility of this work, from being an academic work to a clinically applicable work, could be expanded upon in the discussion. I have two specific routes. First, I think the discussion of the discrepancy to other findings in the literature (paragraph starting line 348) is cursory. I would value expansion upon this consideration as it could relate to the wider use of these data as this process is moved forward into clinical applications. Second, discussion of how to extend this work into a clinical practice remains light. This would be of much greater value if focused onto this in the discussion more.
